# Novel Fluorescent Benzimidazole-Hydrazone-Loaded Micellar Carriers for Controlled Release: Impact on Cell Toxicity, Nuclear and Microtubule Alterations in Breast Cancer Cells

**DOI:** 10.3390/pharmaceutics15061753

**Published:** 2023-06-16

**Authors:** Rayna Bryaskova, Nikolai Georgiev, Nikoleta Philipova, Ventsislav Bakov, Kameliya Anichina, Maria Argirova, Sonia Apostolova, Irina Georgieva, Rumiana Tzoneva

**Affiliations:** 1Department of Polymer Engineering, University of Chemical Technology and Metallurgy, 8 Kliment Ohridsky Str., 1756 Sofia, Bulgaria; 2Department of Organic Synthesis, University of Chemical Technology and Metallurgy, 8 Kliment Ohridsky Blvd., 1756 Sofia, Bulgaria; 3Institute of Organic Chemistry with Centre of Phytochemistry, Bulgarian Academy of Sciences, 1113 Sofia, Bulgaria; 4Institute of Biophysics and Biomedical Engineering, Bulgarian Academy of Science, Acad. G. Bonchev Str., Bl. 21, 1113 Sofia, Bulgaria

**Keywords:** drug carriers, micelles, fluorescence, benzimidazole derivatives, drug release, cytotoxicity, nuclear and microtubule alterations

## Abstract

Fluorescent micellar carriers with controlled release of a novel anticancer drug were developed to enable intracellular imaging and cancer treatment simultaneously. The nanosized fluorescent micellar systems were embedded with a novel anticancer drug via the self-assembling behavior of well-defined block copolymers based on amphiphilic poly(acrylic acid)-block-poly(n-butyl acrylate) (PAA-b-PnBA) copolymer obtained by Atom Transfer Radical Polymerization (ATRP) and hydrophobic anticancer benzimidazole-hydrazone drug (BzH). Through this method, well-defined nanosized fluorescent micelles were obtained consisting of a hydrophilic PAA shell and a hydrophobic PnBA core embedded with the BzH drug due to the hydrophobic interactions, thus reaching very high encapsulation efficiency. The size, morphology, and fluorescent properties of blank and drug-loaded micelles were investigated using dynamic light scattering (DLS), transmission electron microscopy (TEM), and fluorescent spectroscopy, respectively. Additionally, after 72 h of incubation, drug-loaded micelles released 3.25 μM of BzH, which was spectrophotometrically determined. The BzH drug-loaded micelles were found to exhibit enhanced antiproliferative and cytotoxic effects on MDA-MB-231 cells, with long-lasting effects on microtubule organization, with apoptotic alterations and preferential localization in the perinuclear space of cancer cells. In contrast, the antitumor effect of BzH alone or incorporated in micelles on non-cancerous cells MCF-10A was relatively weak.

## 1. Introduction

Cancer is a leading cause of mortality worldwide, responsible for estimated 10 million deaths in 2020, according to the World Health Organization (WHO) [1]. Fundamental discoveries and new technologies in the last decade have led to major advances in understanding the mechanisms of this heterogeneous disease. The traditional chemical drugs for cancer treatment, such as doxorubicin (DOX), camtothecin, and paclitaxel (PTX) for clinical application, are limited due to their severe side effects, low therapeutic efficacy, and serious cytotoxicity [2,3]. Hence, innovative drugs and expensive novel treatments have been developed and applied as targeted therapy. However, progress in cancer treatment is uneven—some malignant tumors respond better, but in others, treatment outcomes are minimal, and the percentage of survival remains very low. Therefore, there is a need to expand the arsenal of novel antineoplastic drugs and develop low-cost targeted drug delivery systems, i.e., by nanocarriers, in order to reduce the side effects of chemotherapeutics. Benzimidazole derivatives play a key role in cancer research due to their anticancer potential with different mechanisms to inhibit tumor progression, minimal toxicity, and facile synthetic strategies allowing a wide variety of structures to be obtained by substitutions at the 1, 2, 5, and/or 6 positions of the benzimidazole core [4,5]. Benzimidazoles that have found application in oncology practice are the alkylating agents from the group of nitrogen mustard analogs, bendamustine [6], the oral mitogen-activated protein kinase inhibitors, binimetinib and selumetinib [7], the poly(ADP-ribose) polymerase (PARP) inhibitor, veliparib [8], the small molecule inhibitor of the sonic hedgehog, glasdegib, which was approved for medical use in the United States in 2018 [9], the multi-target angiokinase inhibitor, dovitinib [10], and the tubulin inhibitors, denibulin, nocodazole, albendazole, mebendazole, etc. [4]. Albendazole and mebendazole have long been applied as antiparasitic agents in clinics, and their pharmacokinetic and pharmacodynamic properties have been thoroughly explored. Moreover, their toxicity on normal human cells and possible adverse effects are well understood. For this reason, in recent years, clinical trials have been conducted to repurpose them as anticancer agents [11]. Inspiringly, these medications do exhibit good antitumor properties with antiproliferative and cytotoxic effects on different cancer cell lines [12,13,14].

However, low water solubility is a limiting factor for the use of benzimidazole-based antineoplastic agents. Drug insolubility causes low in vivo drug concentration and failure to achieve therapeutic effects as well as unforeseeable levels of bioavailability [15]. Compared with conventional medicine, drug delivery systems have several advantages, such as controlled and maintained drug release kinetics, improved patient compliance, easy modification, and low toxicity [16,17].

The issues related to drug delivery can be addressed by adapting to the developmental aspects of nanomedicines, such as formulating them into polymer-based systems such as polymer micelles—self-aggregated colloids made by amphiphilic copolymers. These polymer systems possess unique characteristics with respect to other nanocarriers: smaller size enabling the passive targeting of solid tumors, good solubilization properties of hydrophobic compounds assimilated in the lipophilic core, and a prolonged blood circulation time provided by the hydrophilic corona [18,19]. In addition, polymer micelles ensure high drug loading capacity, low toxicity, and highly efficient drug delivery. Therefore, they appeared as very promising nanosized drug delivery systems and have received great attention in recent years [20,21].

In the literature, there are several studies focused on the development of novel polymeric delivery systems based on natural or synthetic polymers loading benzimidazole derivatives as anticancer drugs, such as albumin nanoparticles [22,23] and polyurethane nanoparticles [24]. Biological tests performed with pure drugs and their polymer nanoformulations revealed a significant anticancer activity on the tested cancer cell lines in favor of nanocarriers [22,23,24]. The developed benzimidazole-loaded polymer nanoparticles possess the potential for reduced frequency of drug administration, lower risk of dose-related adverse effects, and improved bioavailability and efficacy. 

These promising results motivated us to develop novel fluorescent polymer micelles based on amphiphilic poly (acrylic acid)-block-poly (n-butyl acrylate) (PAA-b-PnBA) copolymer loaded with benzimidazole-hydrazone derivative (BzH) drug with proved anticancer properties. It was found that the used BzH showed aggregation-induced fluorescence emission as a result of the loading process. This makes the prepared micelles an excellent tool for theranostic purposes for real-time monitoring of drug delivery. Such theranostic systems provide important information on drug transport and drug distribution in the body, and thereby increasing the quality of medical treatment [25,26,27,28,29,30,31]. The other goal of the present work was to investigate the intracellular distribution of drug-loaded micelles, cytotoxicity, nuclear alterations, and the effect on microtubular organization in cancer and non-cancerous cell lines.

## 2. Materials and Methods

### 2.1. Materials

Tert-butyl acrylate (t-BA) (Alfa Aesar, 99%) and n-butyl acrylate (n-BA) (Alfa Aesar, Ward Hill, MA, USA, 98%) were passed through the neutral Al_2_O_3_ (Fluka, Buchs, Switzerland) column to remove the inhibitor. Copper(I) bromide (Aldrich, San Diego, CA, USA, 98%), ethyl 2-bromopropionate (EBP) (Aldrich, 98%), N, N, N′, N″, N″-pentamethyldiethylenetriamine (PMDETA) (Acros, Geel, Belgium, 98%), butanone (Fisher Chemical, Hampton, NH, USA, 99%), isopropyl alcohol (Macron, Dunkirk, France), and 3,4,5-trimethoxybenzaldehyde, 98% (Alfa-Aesar) and 1,4-dioxane (Merck, Boston, MA, USA, 99%), HCl (Valerus, Sofia, Bulgaria, 36%), dimethylformamide (DMF) (Merck, 99.9%), methanol (Acros), tetrahydrofuran (THF) (Macron) were used as received.

### 2.2. Cell Lines

Human epithelial breast cancer (MDA-MB-231) cell line and non-transformed mammary gland epithelial cell line (MCF-10A) were purchased from the American Type Culture Collection (ATCC, Manassas, VA, USA). MDA-MB-231 cell line was cultured in complete Dulbecco’s modified Eagle’s medium—DMEM (Sigma-Aldrich, St. Louis, MO, USA) supplemented with 10% fetal bovine serum (FBS), L-glutamine (2 mM) and Penicillin (100 U/mL)/Streptomycin (100 µg/mL)/Amphotericin B (0.25 µg/mL). MCF-10A cell line was cultured in Dulbecco’s Modified Eagle Medium/Nutrient Mixture F-12—DMEM F-12 (Sigma-Aldrich, USA), additionally supplemented with insulin (10 µg/mL), hydrocortisone (500 µg/mL), hEGF (20 ng/mL) and cholera toxin (20 ng/mL). Both types of cell lines were grown at 37 °C and 5% CO_2_.

### 2.3. Methods

The hydrodynamic diameter was determined through dynamic light scattering using Malvern Instruments (Zetasizer Nano ZS). The measurements of the hydrodynamic diameter and particle size distribution were performed at room temperature after filtration of the aqueous micellar solution through a 0.45 μm filter. The ultraviolet–visible absorption spectra were recorded on a UV–vis spectrophotometer (ONDA UV-31). Transmission electron microscopy (TEM) analysis was performed with a JEOL 2100 electron microscope at an accelerating voltage of 200 kV equipped with a digital camera. A drop of micellar solution was deposited on a copper grid coated with a carbon film, and the solvent was allowed to evaporate. Absorption spectra were recorded on a Hewlett Packard 8452A spectrophotometer in water. The fluorescent spectra were recorded on a Scinco FS-2 fluorescence spectrophotometer. Thin layer chromatography (TLC) was performed on ALUGRAM SIL G/UV254 pre-coated aluminum sheets with silica gel 60, 0.20 mm thick (Macherey-Nagel, Germany) and eluted by benzene-methanol (4:1, *v*/*v*). The IR spectrum of the synthesized compound in the solid state was recorded on a Bruker Tensor 27 FT spectrometer in ATR (attenuated total reflectance) mode with a diamond crystal accessory at a resolution of 2 cm^−1^, 64 scans. ^1^H and ^13^C NMR spectra were recorded on a Bruker Avance II+ 600 MHz NMR instrument in deuterated-d6 dimethyl sulfoxide (DMSO-d6) as solvent, at room temperature. Chemical shifts (δ) are expressed in parts per million (ppm), and coupling constants (J) are given in Hz. Splitting patterns were indicated by the symbols: s (singlet), d (doublet), t (triplet), and m (multiplet).

### 2.4. Synthesis

#### 2.4.1. Synthesis of ^1^H-Benzimidazole-2-yl Hydrazone Derivative 

Synthesis of ^1^H-benzimidazole-2-yl hydrazine 5 (BzH) derivative was carried out as reported elsewhere [32]. In brief, to a solution of ^1^H-benzimidazol-2-yl-hydrazine (0.002 mol) in absolute ethanol (99%, 5 mL) was added 3,4,5-trimethoxybenzaldehyde (0.002 mol). The reaction was refluxed for 3 h while the reaction was monitored using TLC (benzene/methanol = 4:1, *v*/*v*). The solid product was filtered and recrystallized from ethanol. Yield: 62%; Mp 216.1–218.5 °C; Rf = 0.4; IR (νmax/cm^−1^): 3380 (νN-H), 3060, 3005 (νCHarom), 2940 (νasCH_3_), 2834 (νsCH_3_), 1655 (δNH, νC = N), 1612 (νC = N), 1462 (δasCH_3_), 1359 (δsCH_3_), 1228, 1121 (νC-O-C), and 728 (γC-H); ^1^H NMR (600 MHz, DMSO-d6) δ (ppm): 11.55 (bs, 2H, NH), 7.95 (s, ^1^H, N = CH), 7.27–7.25 (m, 2H, Ar-bz), 7.11 (s, 2H, Ar), 6.99–6.95 (m, 2H, Ar-bz), 3.86 (s, 6H, OCH_3_), and 3.69 (s, 3H, OCH_3_); ^13^C NMR (150 MHz, DMSO-d6) δ (ppm): 153.76, 141.38, 138.76, 131.08, 104.35, 60.58, and 56.52.

#### 2.4.2. Synthesis of Poly (Acrylic Acid)-Block-Poly (n-Butyl Acrylate) (PAA-b-PnBA) Copolymer

Poly (acrylic acid)-block-poly (n-butyl acrylate) copolymer (PAA_40_-b-PnBA_95_) was synthesized according to the procedure reported elsewhere [33]. The PAA_40_-b-PnBA_95_ block copolymer was obtained using a three-step procedure. Briefly, a PtBA macroinitiator (SEC analysis: Mn = 5500 g/mol; PDI = 1.1) was synthesized via ATRP of tBA (17.5 g, 0.14 mol), initiated by ethyl 2-bromopropionate (EBP) (0.5018 g, 2.77 × 10^−3^ mol) in the presence of PMDETA (0.2432 g, 1.40 × 10^−3^ mol)/Cu(I)Br (0.2 g, 1.4 × 10^−3^ mol) catalyst system at 60 °C for 24 h under nitrogen atmosphere. As a solvent, a 2-butanone/isopropyl alcohol mixture (7:3) was used. The PtBA macroinitiator (3.1 g, 0.56 × 10^−3^ mol) was used for synthesis of the PtBA_40_-b-PnBA_95_ (SEC analysis: Mn = 15 300 g/mol; PDI =1.25) copolymer via ATRP in the presence of PMDETA (0.0971 g, 0.56 × 10^−3^ mol)/Cu(I)Br (0.0811 g, 0.56 × 10^−3^ mol) catalyst system at 60 °C for 48 h under nitrogen atmosphere, using nBA (14.0 g, 0.11 mol) as a second block and butanone/isopropyl alcohol (7:3) mixture as a solvent. The composition of the block copolymer PtBA_40_-b-PnBA_95_ was determined by ^1^H NMR as follows: ^1^H NMR of PtBA macroinitiator (400 MHz, CDCl_3_): δ-2.35 ppm (^1^H, –CH– in polymer backbone), 1.9 ppm (2H, –CH_2_– in polymer backbone), 1.46 (9H, –C(CH_3_)_3_). The degree of polymerization (DP = 40) of PtBA was calculated by comparing the peak integrals assigned to the PtBA at 1.46 ppm (9H, PtBA, –C(CH_3_)_3_) and the peak integrals of EBP initiator at 3.77 ppm (^1^H, –CH (Br)– C(C O)–); ^1^H NMR of PtBA-b-PnBA copolymer (400 MHz, CDCl_3_): δ = 4.10 ppm (2H PnBA, –OCH_2_–), δ = 2.24 ppm (^1^H PnBA and ^1^H PtBA, –CH–CO–O–), δ = 1.85 ppm, and hidden under the other signals between 1.2 and 1.7 (2H, PnBA and 2H PtBA, –CH_2_–CH–CO–O–), δ = 1.56 (2H, PnBA, O–CH_2_–CH_2_–CH_2_– CH_3_), δ = 1.46 ppm (9H, PtBA, –C(CH_3_)_3_), δ = 1.35 (m, 2H, PnBA, O–CH_2_–CH_2_–CH_2_–CH_3_), and δ = 0.94 ppm (3H, PnBA, –CH_2_–CH_3_). The degree of polymerization (DPn) of PnBA was calculated by comparing the peak integrals assigned to the PnBA protons at δ = 0.94 ppm (3H, PnBA, –CH_2_–CH_3_) and PtBA proton at δ = 1.46 ppm (9H, PtBA, –C(CH_3_)_3_). Then, PtBA blocks were converted into PAA blocks via hydrolysis with an excess of hydrochloric acid using 1,4-dioxane as a solvent. The comparison of ^1^H NMR (CDCl_3_) spectra of PtBA-b-PnBA and PAA-b-PnBA copolymers showed the complete disappearance of the peak at δ = 1.44 ppm (9H, –C(CH_3_)_3_), characteristic for the methyl groups from the PtBA blocks, which proves the successful hydrolysis [33].

#### 2.4.3. Preparation of Polymeric Micelles

Solution of PAA_40_-b-PnBA_95_ copolymer (20 mg) in DMF (1 mL) was added dropwise at room temperature to distilled water (19 mL) under stirring. The reaction mixture was allowed to stir for 24 h at room temperature. The solution obtained was dialyzed for 24 h in dialysis membrane with frequent changes of deionized water. 

#### 2.4.4. Determination of Critical Micelle Concentration (CMC)

The critical micelle concentration (CMC) of the PAA_40_-b-PnBA_95_ copolymer was determined by the dye solubilization method, as previously reported in [33]. In brief, 20 μL DPH dissolved in methanol (0.4 mM) was added to 2 mL aqueous dispersions of aggregates (eight different concentrations between 1 and 0.001 g/L), and the samples were incubated in the dark for 24 h at 37 °C. CMC value was estimated as an inflection point in the absorbance vs. concentration curve.

#### 2.4.5. Drug Loading

Solution of ^1^H-benzimidazole-2-yl hydrazone derivative 5 (2 mg) and PAA_40_-b-PnBA_95_ copolymer (20 mg) in DMF (1 mL) was added dropwise to distilled water (19 mL) under stirring. The reaction mixture was allowed to stir for 24 h at room temperature. The solution obtained was dialyzed for 24 h in a dialysis membrane ZelluTrans/ROTH T1: MWCO 3500 with frequent changes of deionized water. The micellar solution was filtered (0.45 μm), and the filter was rinsed with DMF. The amount of non-loaded BzH drug in the DMF rinsing fraction was determined by UV spectrophotometry at λ = 340 nm. The encapsulation efficiency (EE) was calculated from the following equations:EE% = (Total mass of drug − Mass of free drug)/(Total mass of drug) × 100(1)

#### 2.4.6. Drug Release

The release test was performed via the dialysis method. An exact volume of micellar solution was placed in a dialysis membrane, which was immersed in buffered media with a pH value of 7.4 at 37 °C. Samples were taken from the external medium, and the concentration of the released drug was determined using the spectrophotometric method (λ = 340 nm) to determine the ^1^H-benzimidazole-2-yl hydrazone derivative release.

### 2.5. Cell Treatment with Micelles and Drug

#### 2.5.1. MTT Test for Cell Viability

To analyze the cytotoxic effect of PAA_40_-b-PnBA_95_ micelles loaded with the drug, an MTT test (Invitrogen, Carlsbad, CA, USA) was performed as previously described [34], with some modifications. The adherent cells were treated with a solution of PAA_40_-b-PnBA_95_ micelles + loaded drug at a concentration of 0.25 mg/mL and with a solution of PAA_40_-b-PnBA_95_ micelles at the same concentration (0.25 mg/mL). Meanwhile, cells were treated with the drug alone in a concentration range of 1 µM to 50 µM in order to calculate the IC_50_ using non-linear regression analysis via GraphPad Prism version 5 software (GraphPad Software, Inc., La Jolla, CA, USA). All samples were further incubated for 24 h and 72 h. As a control, non-treated cells were used. Additional control was added where the cells were treated with DMF at the highest concentration used in the experiments (around 1%) to ensure that the end concentration of DMF that was used to dissolve the drug was not cytotoxic. After the incubation period, the cell medium was changed to a fresh one (100 μL/well). Then, 20 μL of MTT solution (5 mg/mL in PBS) was added. Plates were further incubated for 3 h at 37 °C, and the formed formazan crystals were dissolved by the addition of 100 μL solvent (5% formic acid in 2-propanol) per well and mixing. The absorbance was recorded at 570 nm using a 96-well plate reader Tecan Infinite F200 PRO (Tecan Austria GmbH, Salzburg, Austria). For each concentration, six repeats were performed. A solution of 5% formic acid in 2-propanol (100 μL) was used as a blank. Cell viability (%) was calculated as a percentage of the control value based on the formula: (OD treated/OD control) × 100%.

#### 2.5.2. Fluorescent Imaging

Cells at a density of 4 × 10^4^ cells/well for MDA-MB231 and 3 × 10^4^ cells/well for MCF10-A were seeded on glass slides (d-12 mm, Superior Marienfeld, Lauda-Königshofen, Germany) and placed in 24-well plates. After 24 h of incubation, the cells were treated in the same way as described in the cell viability test and were additionally incubated for 24 h and 72 h. The glass slides were mounted on objective glasses using Mowiol, and the fluorescent images were taken using a spinning disk confocal microscope (Andor Dragonfly 505, Oxford Instruments, Abingdon, UK) with 60× Oil NA 1.4 objective. The obtained data were analyzed by Fiji (1.53t, Bethesda, MD, USA) [35] and compiled via FigureJ plugin [36].

##### Tubulin Immunofluorescence

The cells were fixed with 4% paraformaldehyde, permeabilized with 0.1% Triton X-100, blocked with 1% bovine serum albumin (BSA) dissolved in phosphate-buffered saline (pH 7.4) with 0.1% Tween 20, and labeled with mouse monoclonal antibody against α-tubulin (1:500, Cat. No. MA1-19162, Thermo Scientific, Rockford, IL, USA) for 1 h at room temperature to analyze microtubule organization of both cell lines after the indicated treatment time. A goat anti-mouse secondary antibody conjugated to tetramethylrhodamine (TRITC) (1:1000, Cat. No. sc-2781, Santa Cruz Biotechnology, Dallas, TX, USA) was attached to the primary antibody. The glass slides were mounted on objective glasses using Mowiol^®^4-88 (Cat. No. 81381, Sigma-Aldrich Co. LLC, St. Louis, MO, USA). The fluorescent images were taken using a spinning disk confocal microscope (Andor Dragonfly 505, Oxford Instruments, Abingdon, UK) with 60× Oil NA 1.4 objective with λ_ex_/λ_em_ = 561/617 nm. The obtained data were analyzed by Fiji (1.53t, Bethesda, MD, USA) [35] and compiled via FigureJ plugin [36].

##### DAPI Staining

To observe alterations in cell nuclei, after the cell treatment as described above, they were stained with 1 µg/mL DAPI (Merck, Germany). The glass slides were processed as described above, and the fluorescent images were taken using a spinning disk confocal microscope (Andor Dragonfly 505, Oxford Instruments, Abingdon, UK) with 60× Oil NA 1.4 objective with λ_ex_/λ_em_ = 405/445 nm. Based on cell nuclei staining, cells in mitosis were distinguished, and the mitotic index was calculated using the equation:(2)$$\mathrm{Mitotic}\;\mathrm{index},\;\%=\frac{\mathrm{number}\;\mathrm{of}\;\mathrm{cells}\;\mathrm{in}\;\mathrm{mitosis}}{\mathrm{number}\;\mathrm{of}\;\mathrm{total}\;\mathrm{cells}}\times 100$$

For each condition, at least 5 independent fields of view were used, the mitotic index for each field was determined, and an average of all fields was plotted.

##### Autofluorescence of Micelles with/without Drug

The intensity of autofluorescence of drug-loaded micelles and micelles alone internalized within the cells was measured by a spinning disk confocal microscope (Andor Dragonfly 505, Oxford Instruments, Abingdon, UK) with a 60× Oil NA 1.4 objective, with 405 nm solid-state laser at 15% power, and 600 ms exposure time and gain = 300. MDA-MB-231 and MCF-10A cells were seeded on 35 mm, No 1.5, glass bottom dishes MatTek Corporation (Ashland, MA, USA) at a density of 2 × 10^5^ and 1.4 × 10^5^ cells, respectively. After 24 h incubation at 37 °C, 5% CO_2_, the cells were treated with 0.25 mg/mL of the studied micelles and imaged after 2 h, 24 h, and 72 h. The cells were maintained at 37 °C, 5% CO_2_ during imaging in an Okolab Bold Line Cage CO_2_ incubator (Naples, Italy). The samples were imaged at λ_ex_/λ_em_ = 405/445 nm. At least 5 images were taken for each condition and were further used to calculate the autofluorescence intensity. The acquired images were analyzed via Fiji (1.53t, Bethesda, MD, USA) [35]. 

### 2.6. Data Analysis

The results are expressed as the mean values with standard deviation (±SD) of the indicated numbers of determinations. The statistical significance of differences was determined by analysis of variance (ANOVA) with Tukey’s post hoc test, and differences were considered statistically significant at *p* < 0.05 level. Analyses were performed using GraphPad Prism software version 5 (GraphPad Software, Inc., La Jolla, CA, USA) and Origin(Pro) version 2022 (OriginLab Corporation, Northampton, MA, USA).

## 3. Results and Discussion

Nanosized fluorescent micelles with embedded benzimidazole-hydrazone derivative were obtained using well-defined poly(acrylic acid)-block-poly(n-butyl acrylate) (PAA_40_-b-PnBA_95_) amphiphilic copolymer, which was synthesized by Atom Transfer Radical Polymerization (ATRP) and a novel antitumor agent with pronounced antioxidant and antineoplastic action were developed. 

### 3.1. Synthesis of Benzimidazole-Hydrazone Derivative 5

The benzimidazole-hydrazone derivative-based drug 5 was synthesized in four steps according to Figure 1. First, the ^1^H-benzimidazol-2-yl-thiol 2 was carried out by refluxing ethanol-water solution of potassium hydroxide, carbon disulfide, and o-phenylenediamine using the previously described procedure [32]. Then, ^1^H-benzimidazole-2-thiol was oxidized by a 50% sodium solution of potassium permanganate for 1 h to afford ^1^H-benzimidazol-2-yl-sulfonic acid 3. The filtrate was acidified with hydrochloric acid to pH = 1, and the resulting precipitate of the sulfonic acid was filtered off and washed with water. The ^1^H-benzimidazol-2-yl-sulfonic acid 3 was converted to hydrazine 4 by refluxing it for 3 h in excess of 99% hydrazine hydrate according to the described protocol [37]. Finally, the target compound 5 was synthesized through condensation between ^1^H-benzimidazole-2-yl-hydrazine 4 and 3,4,5-trimethoxybenzaldehyde in a molar ratio 1:1 using 99% ethanol as a solvent. Identification of the ^1^H-benzimidazole-2-yl hydrazone was achieved by spectroscopic techniques, such as FT-IR, ^1^H NMR, and ^13^C NMR.

### 3.2. Preparation of Nanosized Fluorescent Micelles with Embedded Benzimidazole-Hydrazone Derivative 5 (PAA_40_-b-PnBA_95_/BzH)

This was achieved by dissolving PAA_40_-b-PnBA_95_ block copolymer in a good solvent, such as DMF, in the presence of BzH, which was added to an aqueous media, thus leading to the formation of micelles, which consist of hydrophobic PnBA core with included BzH agent and hydrophilic PAA shell according to Figure 2. The established encapsulation efficiency (EE) of the BzH drug was 98% (based on Equation (1)), reaching a final molar concentration of 2.8 × 10^−4^ mol/L.

Initially, the CMC of PAA_40_-b-PnBA_95_ amphiphilic copolymer was estimated. The determination of CMC is of great importance since it provides information for the thermodynamic stability of the micelles or the minimum concentration at which these nanoparticles will stay self-assembled [38]. Therefore, the CMC of PAA_40_-b-PnBA_95_ was achieved by applying the dye solubilization method, using hydrophobic 1,6-diphenyl-1,3,5-hexatriene as a probe, which can be spontaneously solubilized by the hydrophobic cores, giving a characteristic adsorption spectrum with a maximum at 356 nm. By this approach, a CMC of 0.067 mg/mL for PAA_40_-b-PnBA_95_ copolymer was determined (Figure 1), which is in line with our previous results [33]. 

Then, the size and the shape of non-loaded polymer micelles based on PAA_40_-b-PnBA_95_ were determined by dynamic light scattering (DLS) and transmission electron microscopy (TEM). The DLS analysis shows (Figure 2a) that the average hydrodynamic diameter (Dh) of non-loaded PAA_40_-b-PnBA_95_ micelles is 47 nm at relatively narrow polydispersity (PDI = 0.087). In comparison, DLS analysis of the obtained PAA_40_-b-PnBA_95_/BzH micelles (Figure 3a) demonstrates a slight increase in the hydrodynamic diameter, and Dh of 55 nm at PDI = 0.113 is estimated. Further, the CMC of 0.083 mg/mL for PAA_40_-b-PnBA_95_/BzH micelles was determined. The performed TEM analysis reveals the formation of PAA_40_-b-PnBA_95_/BzH micelles with a more contrast hydrophobic core compared with non-loaded PAA_40_-b-PnBA_95_ micelles (Figure 3b). The frequency histogram of the size of the micelles obtained by TEM analysis was performed and compared with the DLS results. The average diameter of the unloaded micelles is 25 nm (Figure 2b), and for drug-loaded micelles, an average diameter of 34 nm is measured (Figure 3b). The difference in the diameters observed by DLS and TEM of unloaded and drug-loaded micelles is a result of the shrinkage of the micelles in their dried state for TEM observation.

The fluorescence emissive properties of the BzH drug and the loaded micelles were further investigated in DMF and aqueous media, respectively (Figure 4). In DMF solution, the BzH compound exhibits a bright fluorescence emission in the UV spectral region between 290 nm and 400 nm with a well-pronounced maximum at 322 nm. In contrast, the prepared nanosized micelles with BzH possessed green fluorescence with a broad spectrum with a peak at 460 nm. The observed red shifting from UV to green region and the broader emission spectrum clearly suggested the incorporation of aggregated benzimidazole-hydrazone into the micelle’s cores. This phenomenon could be rationalized according to Kasha’s exciton theory by the formation of J-aggregates in which the aggregation state of the molecule is regarded as a dipole. That excitonic state of the aggregate was split into two levels through the interaction of transition dipoles [39,40]. It should be pointed out that, unlike BzH-loaded micelles, BzH is non-emissive in an aqueous solution. This was not surprising due to the fluorescence-quenching nature of the solvent molecules.

BzH has a particular water solubility and shows an absorption spectrum in the range of 300–400 nm with a maximum of 340 nm in water (Figure 5a). On the basis of absorption changes at 340 nm, a drug release test was performed via the dialysis method at physiological conditions—in phosphate-buffered saline (pH 7.4) and 37 °C, mimicking the pH in the bloodstream and the cell’s cytosol. The concentration of BzH was determined spectrophotometrically. The observed results presented in Figure 5b reveal a sigmoidal fit (R^2^ = 0.9865) with a fast linear range of the release during the first 24 h, followed by saturation after 48 h. The released content of BzH after 72 h is 3.25 μM.

### 3.3. Cell Treatment with Micelles and Drug

#### 3.3.1. Cytotoxicity Assessment

Cytotoxicity of synthesized BzH drug, micelles, and drug-loaded micelles was performed by MTT test on breast cancer cell line MDA-MB-231 and non-transformed mammary gland epithelial cell line MCF-10A (Figure 6). Firstly, MDA-MB-231 cells were incubated for 24 h and 72 h with the BzH drug in a concentration range from 1 μM to 50 μM (Figure 6a). The estimated IC_50_ values for the BzH drug are >500 μM and 20.3 μM for 24 h and 72 h incubation, respectively. Apparently, the cytotoxic effect of the drug developed over time, and pronounced cytotoxicity is observed at the 72nd hour of incubation. In addition to the direct cytotoxic effect of the drug, a reduced proliferative capacity of MDA-MB-231 cells is also observed, comparing the reduced cell growth between the 24th and 72nd hours of incubation (Figure 6a). Several other studies [22,41,42,43] also confirmed the antiproliferative activity of different benzimidazole derivatives in breast cancer and other cancer cell lines. MCF-10A cells were incubated for 72 h with the BzH drug in the same concentration range to test whether the drug exhibits cell-specific toxicity (Figure 6b).

In the non-cancerous cell line, the cytotoxic effect of the drug turned out to be more than two times weaker, with the calculated IC_50_ value at 51.2 μM (Figure 6b). This fact may play a positive role in avoiding possible side effects if the present benzimidazole-hydrazone derivative is further used in antitumor medical practice. Next, we tested the cytotoxicity of the drug-loaded micelles at different concentrations (0.05 mg/mL to 0.25 mg/mL) on the cancer cell line (Figure 6c). This concentration range was chosen since the stock solution of the micelles is aqueous (1 mg/mL), and in order to treat the cells with it, it is necessary to replace a certain volume of cell medium with the aqueous micelle solution. Thus, we replaced 5, 10, 15, 20, and 25% of the cell medium with an aqueous solution containing the drug-loaded micelles and obtained the above-mentioned concentrations of the drug-loaded micelles. Twenty-five percent seems to be the upper limit for substituting the cell medium with an aqueous solution of the micelles because at thirty percent replacement of the cell medium with water (or 30% dilution of the cell medium with water), the viability of the cultured cells is significantly reduced (Appendix A). As the concentration of the drug-loaded micelles increases, so does their cytotoxicity, reaching up to 50% (IC_50_) when using a 0.25 mg/mL drug-loaded micelle solution (Figure 6c). The efficacy of the drug-loaded micelles to induce a cytotoxic effect on cancer cells is demonstrated by comparing the toxicity (almost 50%) they induced at a concentration of 0.25 mg/mL and the toxicity of empty micelles (C* in Figure 6c) or that of treatment with cell medium replaced by 25% water (C** in Figure 6c), whose toxicity is approximately 10% (Figure 6c). If we assume that a 100% solution of drug-loaded micelles in physiological conditions is able to release 3.25 μM drug content after 72 h of incubation (Figure 5b), a 25% solution of drug-loaded micelles in cell culture would release four times less drug for the same amount of time, or 0.8125 μM, which in turn is able to induce 50% cytotoxicity in cancer cells (Figure 6c). In comparison, using the drug alone required a concentration of 20.3 μM to induce the same 50% cytotoxicity in MDA-MB-231 cells (Figure 6a). Altogether, these results indicate that the drug in its nanoformulation (encapsulated in micelles) appears to be much more effective in inducing an antiproliferative effect in MDA-MB-231 at a lower concentration than that of the drug alone and approaching the IC_50_ value of nocodazole (a well-known anticancer agent that significantly inhibits the polymerization of tubulin), which is 0.76 μM for MDA-MB-231 cells [44]. On the contrary, the same concentration of the drug released from the micelles in MCF-10A cells causes only around 15% toxicity, which is comparable to the toxicity caused by micelle or 25% water in the cell medium (Figure 6d). Confirmation of our work is the study of Racoviceanu et al., who also showed enhanced anticancer properties of albendazole encapsulated in polyurethane structures versus the un-encapsulated compound [24].

#### 3.3.2. Fluorescent Imaging

Benzimidazoles are well known for their effect on tubulin polymerization as they manage to slow down this process [23,45,46]. Therefore, in the following step, we attempted to identify the type of microtubule and nuclear disturbance occurring upon different treatments (Figure 7). The changes in microtubule organization were concentration- and time-dependent. In our study, on the 24th hour, MDA-MB-231 cells treated with the drug alone or with drug-loaded micelles did not show changes in microtubule organization in interphase cells compared to the depolymerized cytoplasmic network in nocodazole-treated cells (Figure 7a). On the other hand, mitotic cells observed in samples treated with the drug, with micelles alone or with drug-loaded micelles, showed some alterations in the spatial organization of cellular microtubules. For example, in telophase, residual interzonal spindle fibers assemble into rod-like structures that persist so that the two daughter cells can separate at the end of mitosis. However, when these fibers persist too long or are thin and short, the division of the daughter cells could be disrupted [47]. Such kind of structures are observed in telophase cells in all treated samples and are more pronounced in cells exposed to the drug alone or to the drug-loaded micelles (Figure 7a, upper row, inset 24 h, circles). No nuclei alterations were found on the 24th hour of treatment in all tested groups, except for nuclei in the two micelle-treated groups, which are swollen to some extent (Figure 7a, middle row). When MDA-MB-231 is treated for 72 h, such rod-like structures of tubulin spindle are observed only in samples with drug-loaded micelles (Figure 7b, insert). In the same samples, the cells exhibit apoptotic topographies, such as shrinkage of the nucleus, fragmented bright nuclei, and condensation of chromatin (Figure 7b, middle row). No other changes in the tubulin network are detected. The presence of microtubule rod-like clusters in daughter cells in samples treated with drug-loaded micelles may be a sign of delayed end-phase mitosis as a result of treatment. When the non-cancerous cell line MCF-10A is treated with 20.3 µM of the drug (IC_50_ for MDA-MB-231 cells), no microtubule changes are observed, and even mitotic cells with well-organized mitotic spindle are visible (Figure 7c, IC_50_**, upper row). Treatment with IC_50_ of the drug for MCF-10A cells (Figure 7c, IC_50_*, upper row) also does not influence microtubule organization. The only observable differences are fewer cells per field of view with a slight elongation. A well-organized tubulin network in interphase cells are observed in samples treated with blank/empty and drug-loaded micelles (Figure 7c, upper row). Regarding the nuclear organization, there are no morphological alterations in the nuclei, implying that DNA was not affected. All this indicates that, in general, the non-cancerous cells do not respond to the treatment. We can summarize that the drug used alone exhibits a short-term transient effect on microtubule organization, while the drug released from the micelles exhibits a long-term effect on microtubule organization, as was reported. Similar results for such effects in tubulin organization were reported in prostate cancer by Chang and colleagues [48].

#### 3.3.3. Mitotic Index

The effect of the drug treatment on the mitotic index was also studied (Figure 8). After 24 h of incubation, the mitotic index is not significantly affected by the different treatments. However, after 72 h of treatment, a steady decrease in dividing cells is observed in the samples containing the drug (Figure 8). These findings are in agreement with the results of the MTT test, which reveals that the cytotoxic effect of the benzimidazole derivative on MDA-MB-231 cells develops over time and appears after 72 h of incubation (Figure 6a). The possible impact of the delayed microtubule organization and apoptotic alterations on the mitotic index as a result of cancer cell treatment with drug alone or with drug-loaded micelles should also be considered (Figure 7).

#### 3.3.4. Fluorescent Intensity

Because the drug-loaded micelles were shown to be fluorescent with maximum intensity at 460 nm, we aimed to follow up on the efficacy and distribution of the drug-loaded micelles in the cells. By detecting the autofluorescence of the drug-loaded micelles, we monitored their gradual accumulation with time in the perinuclear space in both studied cell lines (Figure 9a–d); however, the increase of autofluorescence in MDA-MB-231 cells (Figure 9a,b) after 72 h of incubation is more pronounced compared with that observed for MCF-10A cells (Figure 9c,d). This perhaps explains the presence of swollen nuclei in MDA-MB-231 cells treated with drug-loaded micelles described above (Figure 7a). The obtained results give a reason to assume selectivity towards cancer cells because drug-loaded micelles preferentially accumulate in them over time.

## 4. Conclusions

In summary, new benzimidazole-hydrazone was successfully encapsulated with a very high efficiency of 98% in novel fluorescent micellar carriers consisting of well-defined block copolymers based on amphiphilic poly(acrylic acid)-block-poly(n-butyl acrylate) (PAA-b-PnBA) copolymer. Our in vitro experiments performed on breast cancer cell line MDA-MB-231 and non-transformed mammary gland epithelial cell line MCF-10A highlighted several findings: drug-loaded micelles accumulate in time perinuclearly and preferentially in cancer cells; we revealed higher efficacy of the encapsulated drug (0.8125 μM induce 50% toxicity) to induce cytotoxicity and antiproliferative effect in breast cancer cells than the drug alone (20.3 μM induce 50% toxicity) (Figure 10); a low toxic response from non-cancerous cells; delayed end-phase mitosis by the persistence of microtubule rod-like clusters in telophase daughter cancer cells treated with drug-loaded micelles; and decreased mitotic activity in cancer cells treated with drug-loaded micelles. 

All these findings give us grounds to assume the novel benzimidazole-hydrazone-loaded micelles as an antitumor drug release system with high potential. Further studies should be performed to analyze the cellular mechanisms involved in the cell entry of the micelles, as well as their fate in the cells over a longer period of time in relation to their body clearance in future therapies. Attention should be paid to the mechanism of excretion of loaded micelles from the body because they are not biodegradable and are bigger than kidney glomeruli in order to be excreted by the kidney because the current thinking in bionanotechnology that renal clearance is limited by glomerular basement membrane pore size (≈6 nm). Nevertheless, there is growing evidence existing in the literature confirming that polymer particles exceeding the threshold can also be excreted with urine by an alternative mechanism of renal clearance enabling excretion of polymer particles above glomerular cut-off size, including translocation of the particles through peritubular endothelium to tubular epithelial cells is an alternative mechanism [49,50].

## Data Availability

The authors declare that the data supporting the findings of this study are available within the article.

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
