# Peer review of "Novel Fluorescent Benzimidazole-Hydrazone-Loaded Micellar Carriers for Controlled Release: Impact on Cell Toxicity, Nuclear and Microtubule Alterations in Breast Cancer Cells"

_pharmaceutics, 2023, doi:10.3390/pharmaceutics15061753_

Round 1

Reviewer 1 Report (Previous Reviewer 2)

In the manuscript by Rayna Bryaskova et al. benzimidazole-hydrazone-loaded micellar carriers are developed for controlled release of hydrophobic drug for cancer treatment. This is a resubmission of previously submitted manuscript with many corrections made on the margins. In order to facilitate reading for reviewers, it is recommended to submit clean version without colour indications and comments. I have to say that not all of comments have been addressed in the revised version. As it was said before, the paper can be of interest for the broad readers of Pharmaceutics after several major issues are addressed.

1.       What is the CMC in the presence of benzimidazole-hydrazone drug? It is known that the dye solubilization method for CMC determination is imprecise because hydrophobic dye induces and facilitates micelle formation at lower concentrations. Please compare with other CMC determination methods (ring detachment method, contact angle, etc.).

2.       Why DAPI is cyan? It usually clear blue. What gives red color for the tubulin structures? Please indicate name and excitation and emission wavelengths for red staining tubulin-specific agent.

Some additional points^

1.       What is the pore size of dialysis membrane?

2.       Figure 4b. Is it normalized units? What type of normalization was used?

3.       Is BzH in crystalline form inside the micelles?

4.       Do you have TEM images at higher magnifications?

English is fine

Author Response

Answer to Reviewer 1

1.    What is the CMC in the presence of benzimidazole-hydrazone drug? It is known that the dye solubilization method for CMC determination is imprecise because hydrophobic dye induces and facilitates micelle formation at lower concentrations. Please compare with other CMC determination methods (ring detachment method, contact angle, etc.).
Answer: The CMC of the micelles with loaded benzimidazole-hydrazone drug was determined by dye solubilization method. The obtained value was 0.083 mg/ml, which is closed to the non-loaded micelles. The obtained CMC value was added to the manuscript as suggested by the reviewer. 
The most applied method for CMC determination of polymer micelles is the dye solubilization method, which is very suitable and conventional method because of their high sensitivity (CMC values lower than 0.1 mM) and fast measurement [1,2]. For comparison, the suggested ring detachment method, which is based on measuring the solution surface tension, has some disadvantages as no significant surface tension change observed at concentrations of polymer above the CMC [3], long measurement time and the need of large sample volumes [4].
1. https://doi.org/10.1016/j.jconrel.2021.02.031
2. https://doi.org/10.1016/j.jcis.2005.07.006
3. https://doi.org/10.1016/j.nantod.2012.01.002
4. https://doi.org/10.1007/s10895-018-2209-4

2.       Why DAPI is cyan? It usually clear blue. What gives red color for the tubulin structures? Please indicate name and excitation and emission wavelengths for red staining tubulin-specific agent.
Some additional points
Answer: It seems that we answered to that questions previously but we will do again. Cell nuclei stained with DAPI are only pseudo-colored with Cyan as it provides better contrast to the black background compared to blue color. The excitation and emission wavelengths for DAPI used in this study are the same as the commonly used ones (described in materials and methods part) and are not affected by the pseudo-coloring! Additionally, we consider using Cyan pseudo-colored because cyan (color) is recommended, as it is color-blind friendly. We provide several Ref. to proof that: 
https://www.ncbi.nlm.nih.gov/pmc/articles/PMC6080651/; https://www.ncbi.nlm.nih.gov/pmc/articles/PMC3290635/; https://journals.plos.org/plosbiology/article?id=10.1371/journal.pbio.3001161
Concerning the question” What gives red color for the tubulin structures” we explain as follows: The visualization of tubulin structure is done by immunoflueorescence using primary and secondary antibodies. The tubulin structures are visualized (in red) by fluorophore tetramethylrhodamine (TRITC) which was conjugated to the secondary antibody. Please, see the methodology part. 

3.    What is the pore size of dialysis membrane?
Answer: For the dialysis: the following dialysis membrane ZelluTrans/ROTH T1: MWCO 3500 was used and this information was added to the manuscript in the experimental section.
4.    Figure 4b. Is it normalized units? What type of normalization was used?
Answer: The absorption, fluorescent and excitation spectra have completely different intensity scales with a difference of more than two orders. This makes their direct placement in the same figure irrelevant. That is why for better visualization all spectra were divided by its maximal values, which resulted bands with the same maximal intensity and making comparison between depicted spectra easier for the reader.
5.    Is BzH in crystalline form inside the micelles?
Answer: In our opinion as a result of the obtained data, we conclude that the BzH was included in the core of the micelles not as separated molecule but as an assembly of several molecules, which are in the random architecture. 
6.    Do you have TEM images at higher magnifications?
Answer: There are TEM images at higher magnifications and one of them is inserted in figure 3b.

Reviewer 2 Report (Previous Reviewer 4)

As far as I know, this MS was resubmitted.

The authors made some corrections and added information about the synthesis and structure confirmation of block-copolymer.

However, some of my questions found no answers.

1. What about mechanism of excretion of loaded micelles? The size of 34 nm is too much for the loaded micelles to leave the body through the kidneys (kidney channels are about 3 nm). Copolymer is not biodegradable, so it is unclear. The authors should discuss this.

2. Why only PtBA block hydrolyzed to PAA? Provide the evidence in the MS.

Author Response

Answer to Reviewer 2 
1.    What about mechanism of excretion of loaded micelles? The size of 34 nm is too much for the loaded micelles to leave the body through the kidneys (kidney channels are about 3 nm). Copolymer is not biodegradable, so it is unclear. The authors should discuss this.
Answer: Thanks again for this question to the reviewer. The reviewer is fundamentally correct because the micelles used for drug loading are not biodegradable and the size of the micelles exceeds the threshold of the nephron glomeruli. However, there is increasing evidence in the literature that there is an alternative mechanism of renal clearance by which polymer particles exceeding the threshold can also be excreted in the urine. We discuss this issue in the Conclusion of the document and it will be the subject of our future research: “Attention should be paid about the mechanism of excretion of loaded micelles from the body since they are not biodegradable and are bigger then kidney glomeruli in order to be excreted by the kidney because the current thinking in bionanotechnology is that renal clearance is limited by glomerular basement membrane pore size (≈6 nm). Nevertheless, there is a growing evidences existing in the literature conforming that polymer particles exceeding the threshold can also be excreted with urine by alternative mechanism of renal clearance enabling excretion of polymer particles above glomerular cut-off size including translocation of the particles through peritubular endothelium to tubular epithelial cells is an alternative mechanism [Naumenko, V.; Nikitin, A.; Kapitanova, K.; Melnikov, P.; Vodopyanov, S.; Garanina, A.; Valikhov, M.; Ilyasov, A.; Vishnevskiy, D.;Markov, A.; et al. Intravital Microscopy Reveals a Novel Mechanism of Nanoparticles Excretion in Kidney. J. Control. Release 2019, 307, 368–378; Dogra, P.; Adolphi, N.L.; Wang, Z.; Lin, Y.S.; Butler, K.S.; Durfee, P.N.; Croissant, J.G.; Noureddine, A.; Coker, E.N.; Bearer, E.L.; et al. Establishing the Effects of Mesoporous Silica Nanoparticle Properties on In Vivo Disposition Using Imaging-based Pharmacokinetics. Nat. Commun. 2018, 9, 4551].

2. Why only PtBA block hydrolyzed to PAA? Provide the evidence in the MS.
Answer: In order to obtained amphiphilic block copolymer based on PAA-b-PnBA, it was applied the acid-catalyzed elimination of the tert-butyl groups from PtBA-b-PnBA block copolymer by which only hydrolysis of PtBA block occurred. 
In the manuscript in the Experimental section was added 1HNMR data for the hydrolysis of PtBA-b-PnBA copolymer to PAA-b-PnBA copolymer as follow: “The comparison of 1H NMR (CDCl3) spectra of PtBA-b-PnBA and PAA-b-PnBA copolymers showed the complete disappearance of the peak at δ = 1.44 ppm (9Н, –C(CH3)3), characteristic for the methyl groups from the PtBA blocks, which proves the successful hydrolysis.”

Round 2

Reviewer 1 Report (Previous Reviewer 2)

The paper is now suitable for publishing in Pharmaceutics. Recommendation accept. 

English is readable/

This manuscript is a resubmission of an earlier submission. The following is a list of the peer review reports and author responses from that submission.

Round 1

Reviewer 1 Report

Please comment to the author on the percentage of similarity with the ithenticate program, to avoid problems of duplicating information and to change all possible coincidences.

1.- Improve the abstract since it is a very important topic and at first it is somewhat weak, in addition to defining if the method has already been carried out: self-assembly of block copolymers.

2- In the introduction, add more information on the release method and what type of drug loaders are being used.

3.- better describe the methodology.

4. Line 170 add 1H NMR results and discuss with what is reported in the literature.

5. Figure 2 and Figure 3 in the TEM perform the frequency histogram of the size of micelles and compare with the DLS results.

6. Line 370-374 BZH adsorption changes from 322 nm to 340 nm why? Discuss with what is reported in the literature,

7.- Figure 8 after 72 hr there is a decrease in the mitotic index due to which, discuss and add references.

8.- It is necessary to add studies of release kinetics, in how long the drug is released.

9.- Add the drug release mechanism to create a scheme.

10.- Conclusions, based on the cytotoxicity results, it is necessary to add which is the concentration of the drug with the best results for cell viability.

Reviewer 2 Report

In the manuscript by Rayna Bryaskova et al. benzimidazole-hydrazone-loaded micellar carriers are developed for controlled release of hydrophobic drug for cancer treatment. The paper can be of interest for the broad readers of Pharmaceutics after several major issues are addressed.

1.       Please explain what the sonic hedgehog is (line 54). Explanation should be given to be clear for the broad readers of Pharmaceutics.

2.       What is the CMC for PAA40-b-PnBA95 amphiphilic copolymer, which is different from classic surfactant micelles as supposed? How does the polymer exist before and after CMC in contrast to monomers, premicellar aggregates and micelles for surfactants?

3.       What is the CMC in the presence of benzimidazole-hydrazone drug? It is known that the dye solubilization method for CMC determination is imprecise because hydrophobic dye induces and facilitates micelle formation at lower concentrations. Please compare with other CMC determination methods (ring detachment method, contact angle, etc.).

4.       Please supplement Figure 4 with corresponding excitation and absorption spectra.

5.       460 nm belongs to blue region. Please revise line 359.

6.       Why DAPI is cyan? It usually clear blue. What gives red color for the tubulin structures? Please indicate name and excitation and emission wavelengths for red staining tubulin-specific agent.

Reviewer 3 Report

In this work, the authors described the BzH drug-loaded micelles exhibiting enhanced antriproliferative and cytotoxic effects on MDA-MB-231 cells and long-lasting effects on microtubule organization. Some specific issues should be addressed before further considerations. The authors should provide a more intuitive comparison of PAA40-b-PnBA95/BzH and BzH. Unfortunately, no significant advantage of cell viability was found comparing PAA40-b-PnBA95/BzH and BzH and so the innovation of this work should be improved.

The manuscript need to be checked and revised carefully in format. For example, the title had better be one sentence rather than two. The horizontal coordinate of Figure 5(b) should be given. The font of figure 6c and 6d should be the same, Line 114, CO2 should be CO2; Line 162, 10-3 should be 10-3 and so on. According to fig. 3b, the uniformity of nanoparticles doesn't seem to match the PDI of DLS. A new fig.3a with a larger size range and new TEM should be provided.

Reviewer 4 Report

The manuscript by Bryaskova et al. is devoted to the development of a benzimidazole derivative delivery system based on self-associating micelles of a acrylic acid-b-butyl acrylate copolymer. The efficiency of drug encapsulation (an extremely high value of 98%), micelle size, cytotoxicity, tubulin immunostaining, and DAPI staining were evaluated in the work. The work is of interest, but there are a number of serious comments to the authors.

1. What determines the choice of the ligand-catalyst ratio in ATRP? What conversion of monomers does it lead to? It is also necessary to provide a more detailed description of the synthesis of the block copolymer, incl. structure confirmation.

2. After the synthesis of acrylic acid-b-butyl acrylate copolymer, the product was modified by hydrolysis. At the same time, the authors stated that only poly-tert-butyl acrylate units undergo hydrolysis. Why?

3. What do the numbers 40 and 95 mean in the name of the PAA40-b-PnBA90 copolymer?

4. It is not clear how the authors determined the encapsulation efficiency. From the total amount of the drug was subtracted unbound, determined spectrophotometrically? Is the amount of DMF used excessive (1 ml for 2 and 20 mg)?

5. Given the low solubility of the benzimidazole derivative in water (this can be seen from the spectrum), is it correct to study the release of the drug using UV detection?

6. The size of drug-loaded micelles is more than 50 nm. How, according to the authors, should they be excreted from the body: the polymers used are not biodegradable, and the size of the renal tubules is about 3 nm?

7. Typos in the text: lines 133, 148: typos “cm-1”, lines 162, 163, 167: typos “10−3”